# An Efficient Feature Selection Algorithm for Gene Families Using NMF and ReliefF

**DOI:** 10.3390/genes14020421

**Published:** 2023-02-06

**Authors:** Kai Liu, Qi Chen, Guo-Hua Huang

**Affiliations:** 1College of Plant Protection, Hunan Agricultural University, Changsha 410128, China; 2Hunan Provincial Key Laboratory for Biology and Control of Plant Diseases and Insect Pests, Hunan Agricultural University, Nongda Road, Furong District, Changsha 410128, China; 3College of Information and Intelligence, Hunan Agricultural University, Changsha 410128, China

**Keywords:** gene family, NMF-ReliefF, feature selection, classification, insect genome

## Abstract

Gene families, which are parts of a genome’s information storage hierarchy, play a significant role in the development and diversity of multicellular organisms. Several studies have focused on the characteristics of gene families, such as function, homology, or phenotype. However, statistical and correlation analyses on the distribution of gene family members in the genome have yet to be conducted. Here, a novel framework incorporating gene family analysis and genome selection based on NMF-ReliefF is reported. Specifically, the proposed method starts by obtaining gene families from the TreeFam database and determining the number of gene families within the feature matrix. Then, NMF-ReliefF is used to select features from the gene feature matrix, which is a new feature selection algorithm that overcomes the inefficiencies of traditional methods. Finally, a support vector machine is utilized to classify the acquired features. The results show that the framework achieved an accuracy of 89.1% and an AUC of 0.919 on the insect genome test set. We also employed four microarray gene data sets to evaluate the performance of the NMF-ReliefF algorithm. The outcomes show that the proposed method may strike a delicate balance between robustness and discrimination. Additionally, the proposed method’s categorization is superior to state-of-the-art feature selection approaches.

## 1. Introduction

Gene families are groups of genes that have evolved from a common ancestor, and share similar sequences and functions [1]. They are a crucial aspect of genetics and genomics, and their study can provide valuable insights into the evolution, function, and regulation of genes [2]. Additionally, they are enormous units of information and estimation of genetics, contributing significantly to the development and diversity of multicellular organisms. They are also integral to the genomic information storage hierarchy [3]. In evolution, the expansion and contraction of gene families is caused by various factors, including natural selection, genetic drift, and gene duplication. The adaptive gene family expansion occurs when natural selection favors more gene copies [4]. On the other hand, genetic drift can lead to the contraction of a gene family over time due to random changes in the frequencies of the genes in the population. The accumulation of loss-of-function mutations frequently leads to adaptive shrinkage of gene families [5]. Environmental factors are also responsible for gene loss [6]. Gene duplication, where a gene is copied, and the copies are free to evolve independently, can also lead to the expansion of a gene family. When a nonsense mutation stops gene transcription prematurely, it becomes permanent in the population, resulting in its loss.

Due to variances in gene acquisition and loss rates, the copy number of homologous gene families varies significantly among species. It is well known that gene copy number variation can be responsible for the phenotypic novelties of particular species. For example, there are now several ways of identifying insect eating patterns [7,8,9]. A recent study found that human physical features may be predicted using whole-genome sequencing data [10]. However, there has been no advancement in the method of analysis from the standpoint of the gene family. Therefore, we concentrated on extracting species traits at the gene family level to achieve this goal.

Ortholog databases are intensively used to analyze species traits at the gene family level. The orthodox dichotomy has proved useful, although it has inherent limitations [11]. Commonly-used databases include OMA [12], OrthoDB [13], TreeFam [14], and eggNOG [15]. In principle, tree-based methods are preferable because they involve explicit evolutionary models that allow the classification of orthologs, co-orthologs, in-paralogs, and out-paralogs [16]. TreeFam, which belongs to the tree-based method, has fewer erroneously assigned genes than the above database [17]. Here, we defined gene families with the TreeFam tool. TreeFam is a database of phylogenetic trees of gene families identified from animal genomes. It aims to establish a curated resource that provides reliable information on ortholog and paralog assignments and the evolutionary history of gene families. Curated families are introduced in stages, similar to Pfam, based on seed alignments and trees. TreeFam provides curated trees for 690 families and automatically produces trees for an additional 11,646 families. These comprise about 128,000 genes from nine fully sequenced animal genomes and over 45,000 more animal proteins from UniPort [18]; around 40–85 percent of proteins are encoded from fully sequenced animal genomes. The seed families for TreeFam-B are taken from PhIGs clusters. They are expanded by a seed-to-full procedure to form whole families. Manual curation makes TreeFam-B families become TreeFam-A families, which can also be curated later.

Treefam calculations yielded the distribution of gene family members. However, these statistics have a high dimension, with too many gene families and a limited number of species, making it challenging to find meaningful patterns. Feature selection algorithms are often employed to reduce dimensionality to solve dimensional disasters. The reduction of feature dimensionality is a fundamental principle of classification, which primarily attempts to characterize the data set more accurately. It is accomplished by removing the data set’s unneeded, undesirable, and irrelevant characteristics. The most commonly used dimensionality reduction algorithms at the moment are genetic algorithm (GA) [19], random forest (RF) [20], clustering analysis (CA) [21], relief series algorithm (RSA) [22], principal component analysis (PCA) [23], and so on. Further, to solve the multi-classification problem, the ReliefF algorithm is proposed [24]. ReliefF is one of the most significant algorithms utilized in various financial applications. Recursive feature elimination (RFE), one of the most popular feature selection approaches, is effective in data dimension reduction and efficiency increase [25]. Recent studies have shown that consensus-guided unsupervised feature selection (CGUFS) performs well in feature selection for identifying disease-associated genes [26]. Nonnegative matrix factorization (NMF) has been shown to perform well in analyzing omics data. NMF assumes that the expression level of one gene is a linear additive composition of metagenes. The elements in the metagene matrix represent the regulation effects and are restricted to non-negativity [27]. We created an entirely new classification approach in this research that is based on the well-known ReliefF algorithm and nonnegative matrix decomposition (NMF) [28]. This work evaluates the performance of the proposed technique using four publicly available microarray data sets, each containing a large number of cases. The findings reveal that the proposed technique has superior performance in terms of processing time and memory requirements to a variety of mainstream classification methods.

Here, the feature extraction process was illustrated using the insect genome as an example. We conducted data mining on insect gene families and examined the relationship between insect feeding and gene families. Insect feeding habits are dietary preferences acquired by insects throughout the long evolutionary process. Insect survival and reproduction are dependent on feeding choices. Various types and dietary ranges of insects exist, and the same species have different eating behaviors. Herbivorous, carnivorous, sacrificial, and omnivorous are insects’ dietary classifications [29]. Hundreds of insect genomes have been sequenced as whole-genome sequencing costs have been reduced drastically [30]. Several gene families have been linked to energy function in comparative genomics. The framework we developed to build an extremely accurate predictive classifier also considered these gene families as potential characteristics. Finally, we demonstrate a novel genetic method for analyzing the feeding habits of different species of insects, a method that could also be applied to other biological groups.

## 2. Materials and Methods

We first downloaded and selected genome sequences containing the high-quality annotation file. The longest sequence length in the genomic mRNA sequence is retained and the rest of the alternative splicing is removed. Then, using the TreeFam software and its database, we categorize the genome sequence of each species and construct a script to count the classification results. Here, we design and implement a novel feature selection algorithm, NMF-ReliefF. At last, the final classification model is obtained by training the classifier on the reduced dimensionality feature matrix. Figure 1 illustrates the workflow for the proposed predicting methods. The proposed framework of this research work was done using MATLAB of version R2018a. The computer’s CPU is an Intel i5 dual-core 8400H with a primary frequency of 2.80 GHz, and the memory size is 8 GB.

### 2.1. Genome Resources and Species Selection

We downloaded 139 genome sequences with coding gene annotation files, including Coleoptera, Diptera, Hemiptera, Hymenoptera, and Lepidoptera, from the National Center for Biotechnology Information [31], InsectBase [32], VectorBase [33], Fireflybase [34], Ensembl Genomes [35], and GigaDB [36] to allow for more in-depth analysis (Appendix A). The corresponding coding genes had to be found based on the annotation file and the gene sequencing data. We filtered out species with low-quality genomes using the Scaffold N50 genome characteristic value, which is positively related to genome quality, and the more significant, the better. Species with scaffold N50 < 400 Kb genomic assemblies were eliminated. The most extended transcript was chosen when there were many alternative splicing variants for a protein-coding gene. We selected 50 insect species containing the annotation file, 27 of which were verified by literature references as herbivorous and used as positive samples. Twenty-three insect species have been shown in the literature not to feed mainly on plants. Therefore, they are used as examples of non-herbivorous insects. (Appendix A).

### 2.2. Gene Family Analysis

From the alternative splicing file, the genomic mRNA sequences are retrieved first. Alternative splicing generates several RNAs from the sequences of mRNA in the genetic material. Alternative splicing is a biological process in which exons from the same gene are connected in various ways, producing unique but related mRNA transcripts. Alternative splicing causes a gene to produce several mRNAs, which, if left untreated and processed using the TreeFam database, can substantially bias the results. We consequently retained the longest mRNA sequence. TreeFam, which considers phylogenetic relationships, was used to identify gene families derived from a single gene of the most recent common ancestor. The TreeFam script and the TreeFam-A database determined the number of each species’ mRNA sequences corresponding to each TreeFam gene family. A numerical matrix comprised the final configuration.

### 2.3. Feature Selection

Several approaches to feature selection have been applied in bioinformatics. In this paper, we compare our proposed method with three widely used feature selection approaches: support vector machine recursive feature elimination (SVM-RFE) [37], ReliefF [38], and PCA-ReliefF [39]. The SVM-RFE approach for gene selection was created by integrating a minimum-redundancy, maximum-relevancy (MRMR) filter. The mutual information among genes and class labels is used to determine the relevance of a collection of genes, and the mutual information among the genes is used to determine redundancy. Because it considers gene redundancy during gene selection, the technique enhanced the detection of cancer tissues from benign tissues on numerous benchmark data sets. On most data sets, the approach chose fewer genes than MRMR or SVM-RFE. Gene ontology analyses revealed that the method selected genes that are relevant for distinguishing cancerous samples and have similar functional properties.

The Relief method is a feature-weighting technique developed by Kira that applies varying weights to characteristics based on the association between each feature and category [38]. Features having less than a specific weight will be eliminated. The Relief algorithm’s association between features and categories is based on the features’ capacity to discern nearby samples. Relief algorithms are practical and generic attribute estimators. They can discover conditional relationships and give a unified picture of attribute estimates in regression and classification. Furthermore, their quality estimations have a natural meaning. The running time of the Relief algorithm rises linearly with the number of samples m and the number of original features N, resulting in excellent running efficiency. In the Relief series algorithm, k closest neighbors (near misses) are identified, and each feature is given a weighted value. It is a feature-weighting algorithm that is efficient and does not have a data type restriction. Due to the algorithm’s preference for highly relevant features, this algorithm cannot effectively eliminate redundant features.

PCA is a practical approach to optimize variance in each direction and reduce correlations in training data. However, it only helps classification systems indirectly. ReliefF can score each feature’s contribution and offer intuitive evidence by linking the feature and classification accuracy, but correlations between features diminish performance, especially when the features are essential. Zeng et al. [39] first retrieved Mel Frequency Cepstral Coefficient features. A feature selection approach based on PCA and ReliefF is presented to choose the most discriminatory group of features. 

Inspired by the above method, we designed a new feature selection method based on NMF-ReliefF. Given a nonnegative observation data matrix m×n, each column of denotes a sample vector, m represents the number of features, and n represents the number of samples. The NMF algorithm aims to seek two nonnegative matrices, *W* and *H*, which can well reconstruct the matrix as follows:(1)V≈WH

The squared Euclidean distance is the commonly used cost function to measure the quality of the approximation which can be written as follows:(2)$$\min{\left\|V-WH\right\|}_{F}^{2}\;s.t.W\geq 0,H\geq 0$$
where ${\left\|•\right\|}_{F}$ stands for the matrix Frobenius norm. By adopting multiplicative update rules for nonnegative optimization [40], the updating rules of (2) can be obtained as follows:(3)$$W=W⊙\frac{XH^{T}}{WHH^{T}}$$
(4)$$H=H⊙\frac{W^{T}X^{}}{W^{T}WH^{}}$$
where ⊙ shows the Hadamard product, and denotes the transpose of the matrix.

Algorithm 1 shows the iterative algorithm for learning an NMF decomposition and ReliefF feature selection [41], where the multiplicative update rules are given in matrix notation. The operator · denotes pointwise multiplication and the operator/pointwise division.

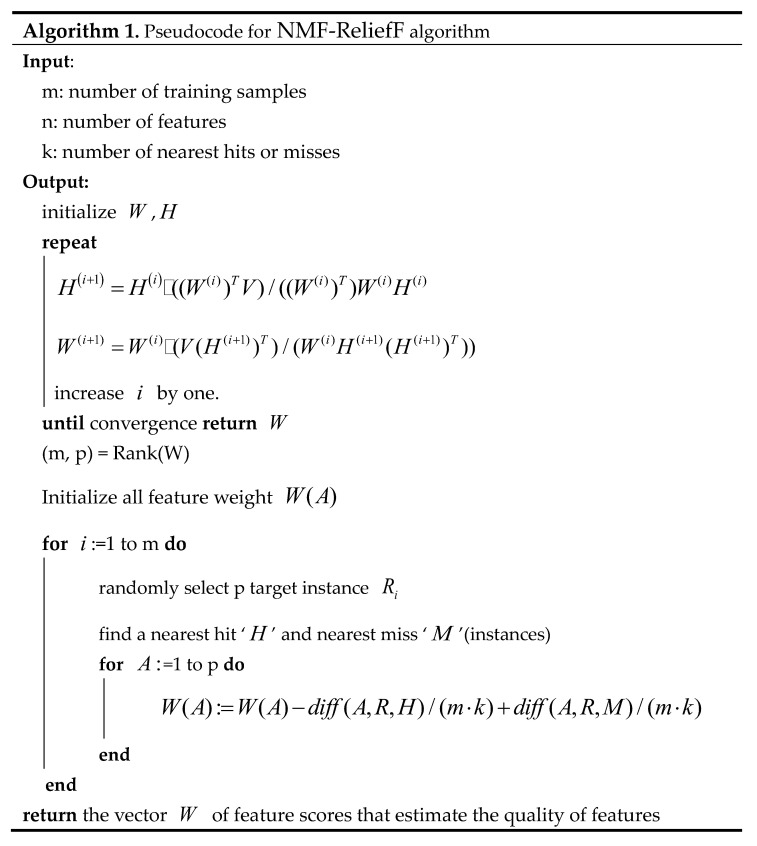


### 2.4. Classification Methods

This research employs three classification methods: Support Vector Machine, Random Forest, and k-Nearest Neighbor, to examine the selected gene subset for categorization of microarray data.

Support Vector Machines (SVM) are supervised learning methods for analyzing data for classification and regression analysis [42]. The SVM training technique results in the assignment of new instances to one of two categories, creating a binary linear classifier that is non-probabilistic. The SVM model represents instances as points in space using Platt scaling. However, it can also be used in probabilistic classification scenarios. They are mapped so that as much distance as possible separates the examples of distinct categories from each other. In the next step, new examples are mapped into this space and their membership in a given category is determined based on where they fall within the gap.

As an ensemble learning method, random forests can perform classification, regression, and other tasks by constructing a large number of decision trees at training time. This is done by identifying the class representing the mean prediction of all the individual trees in a given category. Using random decision forests, overfitting training sets can be corrected with Random Forests (RF).

K-Nearest Neighbor (k-NN) is a non-parametric classification and regression technique [43]. Input consists of the k nearest training examples in the feature space. k-NN assigns items to the category with the highest frequency among its k nearest neighbors based on the majority vote of its neighbors (k is a positive integer, usually a decimal number). The attribute value of the item, the weighted average of the importance of its k nearest neighbors, is the result of a k-NN regression.

To eliminate “selection bias,” we utilize five-fold cross validation (CV) [44] in our studies on each microarray data set with a specified gene subset for each classification technique. To prevent selection bias, we employed the fivefold crossover method to test three classifiers on the previous stage’s data produced from feature selection. Specifically, the data were randomly divided into five sections, of which one copy was used for training and the other was used for testing. This procedure is repeated, with each copy serving as a test set.

### 2.5. Prediction Accuracy Assessment

The prediction accuracy (*ACC*), the area under curve (*AUC*), sensitivity (*SEN*), and specificity (*SPE*) are utilized in this study to assess the effectiveness of various approaches. Their definitions may be found below. The receiver operating characteristic curve (ROC) and area under the ROC curve (*AUC*) demonstrate the detailed performance of various approaches. The ROC curve’s *X*-axis represents the false positive rate (*FPR = 1 − SPE*), while the *Y*-axis represents the true positive rate (*TPR = SEN*). The models in this study are evaluated and compared using five-fold cross-validation.
(5)$$ACC=\frac{TP+TN}{TP+TN+FN+FP}$$
(6)$$SEN=\frac{TP}{TP+FP}$$
(7)$$SPE=\frac{TP}{TP+FN}$$
(8)$$AUC=\frac{\sum pred_{pos}>\sum pred_{neg}}{positiveNum*negativeNum}$$

## 3. Results and Discussion

### 3.1. Data Sets

Four publicly available microarray data sets were used to test the effectiveness of the proposed gene selection method. For better performance and evaluation of the proposed method, we chose the cancer microarray data set, which contains only two classes and is widely used in related work [45,46,47]. These data sets are collected to diagnose various cancers such as prostate cancer, breast cancer, lung cancer, and myeloma. All four microarray data sets share the following characteristics: (1) they are typically high-dimensional, and three exceed 10,000 dimensions. (2) There are fewer than 200 samples, much fewer than the genes. (3) Many redundant and irrelevant genes in these data sets affect classification. The statistics of these data sets are summarized in Table 1.

### 3.2. The Selection of Classifier

We examined three popular classifiers: closest neighbor (k-NN), support vector machine (SVM), and random forest (RF). We evaluated the efficacy of our applied classifier by looking at how well it performed under the proposed scheme. To create a baseline model, we do not use feature selection methods but all features directly. NMF-ReliefF was used to pick features in the proposed method’s preprocessing stage. Table 2 compares three classifiers based on an evaluation of their prediction accuracy. The evaluation of prediction accuracy reveals that our categorization performance is exceptional. The *AUC* values for SVM, RF, and k-NN classifiers are 0.843, 0.723, and 0.745, respectively. The SVM classifier has a significantly higher *AUC* than other classifiers. In addition, the computational times for SVM, RF, and k-NN classifiers are 0.089 s, 0.110 s, 0.122 s, and 0.095 s, respectively. The temporal efficiency of the SVM classifier is superior to that of other classes. In summary, the SVM classifier space is preferable to other examined spaces based on classification performance and average execution time.

### 3.3. Classifying Insect Feeding Habits by Machine Learning

The results indicated that the method performs well in classifying insects as herbivorous, with an average accuracy of 84.5%. Sensitivity, specificity, and the *AUC* (Area under the Curve of ROC) were 84.3%, 87.4%, and 91.9%, respectively, suggesting good performance by the classifier (Table 3). From the results in the table, our designed algorithm achieves better classification results with a classification accuracy of 84% and time consumption of 1.3224, which is significantly higher than other algorithms (Appendix A). It does not take the least amount of time, but compared to the PCA-ReliefF algorithm, it improves accuracy by about 18%, which is better than most other algorithms.

### 3.4. Feature Selected Reflect the Relationship of Gene Family

To illustrate why PCA is inferior to NMF, we extract features and construct a heat map in Figure 2 and Figure 3. While Figure 2 demonstrates that there is no noticeable difference in the average values of herbivorous and non-herbivorous insects, Figure 3 illustrates the opposite. This demonstrates that the NMF method is superior to the PCA algorithm in selecting features in this context.

As shown in Figure 3, our method is effective because the feature heat maps of the screened gene families show discernible differences. These screened features are closely related to insect feeding habits, which can be crucial in building a classifier. As long as the appropriate classifier is selected, the insect’s genes can determine whether it is herbivorous. Since it differs from the traditional homology alignment method, we cannot explain the specific gene family effects with the TreeFam method. Nonetheless, the Pfam database contains some associations.

### 3.5. Comparison with Other Gene Selection Methods

Similar to the classifier mentioned above, four publicly accessible microarray data sets were utilized to examine the efficacy of gene signature selection methods for the specified characteristics to evaluate classification results. This study compared four commonly used feature selection methods, ReliefF, SVM-REF, PCA-ReliefF and NMF-ReliefF. 

Table 2 demonstrates that (1) when the NMF-ReliefF process extracts the features, the classification *ACC* achieved with the SVM classifier ranges from 85 to 95 percent, depending on the data set. (2) The data set’s positive and negative case preferences have a more significant influence on the categorization. In comparison with Table 2, Table 3 shows a 15% increase in *ACC*. (3) The SVM classifier’s classification performance is much superior to that of the k-NN and RF classifiers, as demonstrated by the benchmark test. (4) NMF is marginally superior to PCA for feature extraction, with classification results for the data set indicating an improvement of between 3 and 5%.

We compared our technique to the most cutting-edge algorithms using the test data. We investigated four high-dimensional microarray data sets and calculated the mean and standard deviation for each microarray data set’s accuracy, specificity, sensitivity, and area under the curve. The comparison results are displayed in Table 2. Our approach has a mean precision of 91.3%, a sensitivity of 86.5%, a specificity of 93.2%, and an area under the curve of 88.4%. Our method outperforms ReliefF and PCA-ReliefF in terms of precision, specificity, sensitivity, and extent under the curve. In addition, we have developed a considerably improved approach than SVM-REF.

### 3.6. The Relationships of Selected Features

We selected 50 significant features using the NMF-ReliefF feature selection method, calculated the Pearson correlation coefficient between any two features, and used these results to create heat maps. The feature correlation heat map illustrates the linear correlation between each feature. Different features represent different numbers of gene families, and Figure 4 illustrates that these components are correlated. The coefficients between the coefficient matrices have considerable weight and play a key role in feature selection, allowing us to recognize the corresponding gene family as having a pivotal role. The heat map shows that the critical coefficients are crucial in how a species feeds. We can analyze the gene families associated with these critical coefficients to understand how they work.

### 3.7. Classification Performance with Different Numbers of Selected Gene Families

To determine the ideal number of selected genes, we tested the classification ability of multiple approaches employing varying numbers of selected genes. Note that the *n* option modifies the NMF-ReliefF feature count. In the experiments, the range of *n* values was 5, 10, 20, 30, 40, and 50, while all other parameters remained unchanged. For different feature selection algorithms and parameters, a five-fold cross-check is performed and the run is repeated 15 times to obtain the mean and standard error. Figure 5 depicts the *ACC* curves of the four feature selection techniques with varying feature counts. The results show that if the number of features is less than 10, the *ACC* of the classification evaluation index is less than 75%. However, suppose the number of features is more than 10. In that case, the evaluation index *ACC* can reach 85%, indicating that if the number of features is too small, the classifier is underfitted and cannot provide better classification. In contrast, the evaluation index *ACC* for more than 30 features is stable at about 80%, with a slowly decreasing trend as the number of features increases. According to the experimental statistical results, the *ACC* values for feature numbers 5, 10, 20, 30, 40, and 50 are 0.6709, 0.8915, 0.8438, 0.8418, 0.8276, and 0.8124, respectively. The *ACC* value for a feature count of 10 is much greater than the *ACC* for a lower feature count, while the distance is smaller than the *ACC* for a higher feature count. Therefore, when the number of characteristics is between 10 and 20, our classification approach performs better and does not have too many features to influence the subsequent analysis (Appendix A). This algorithm achieves a better balance between robustness and differentiation than the other algorithms in every case involving an eigenvalue, as shown in Figure 5. The corresponding values for each algorithm, however, are quite high. This results in no significant differences in sensitivity between the algorithms when the number of characteristics chosen is taken into account. In contrast, the specificity varies by more than 15%, with a maximum of approximately 40 unique features. In light of the low sensitivity, the selected features increase as the number of selected features increases. The reason for this is that the number of features available increases as well. Therefore, it is appropriate to take more features when selecting the number of features, even if accuracy is consistent.

## 4. Conclusions

This paper proposes a framework for intrinsically mining associations in gene family data sets and a novel feature selection method based on NMF and ReliefF. The framework can classify feature attributes and is applied to the gene family feature map of insects, which has a good classification ability for insect predators. Furthermore, our proposed feature selection method, NMF-ReliefF, can effectively improve the classification ability in the case of high dimensionality and small data samples. Validation of the algorithm on four publicly available microarray data sets illustrates the effectiveness and superiority of the algorithm, showing that our classification system outperforms most comparable algorithms. Further, it was compared in terms of temporal performance, outperforming most dimensionality reduction-based methods. In the future, we will further analyze the genetic and intrinsic association between the multiclassification performance of the feature selection algorithm and the selected gene families.

## Figures and Tables

**Figure 1 genes-14-00421-f001:**
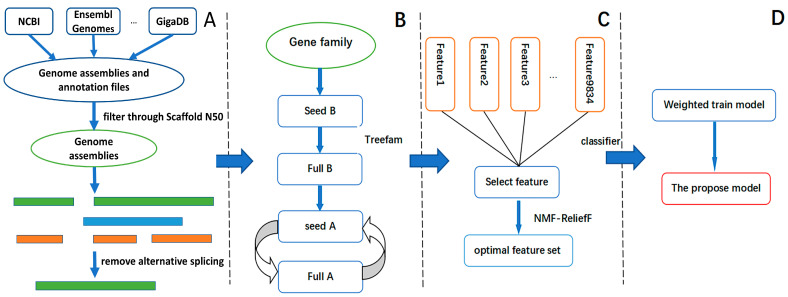
The framework of feature selection algorithm on gene families. Our method consists of four modules. (**A**) We collected genomes with annotation files from individual genomic databases filtered by Scaffold N50. The longest sequence length in the genomic mRNA sequence is retained and the rest of the alternative splicing is removed. (**B**) Using the TreeFam software and its database, we categorize the genome sequence of each species and construct a script to count the classification results. (**C**) Here, we design and implement a novel feature selection algorithm, NMF-ReliefF. (**D**) The final classification model is obtained by training the classifier on the reduced dimensionality feature matrix.

**Figure 2 genes-14-00421-f002:**
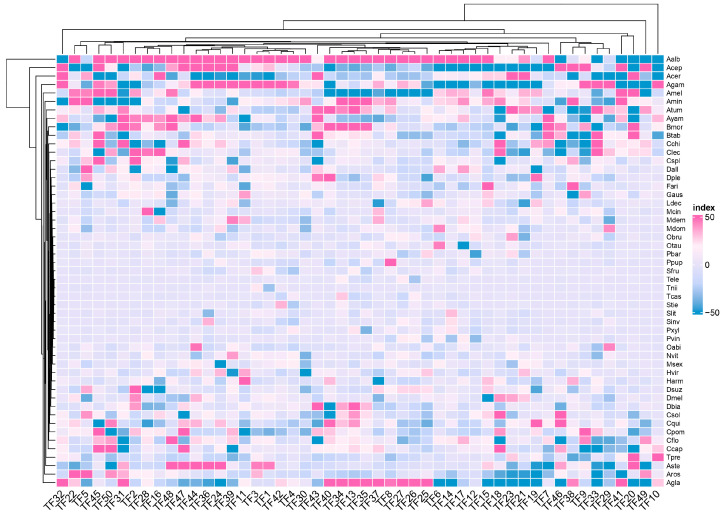
Characteristic heat map of PCA, the vertical axis is the name of insects, and the horizontal axis is the eigenvalues of different features. See Appendix A for the names of species assigned to each label.

**Figure 3 genes-14-00421-f003:**
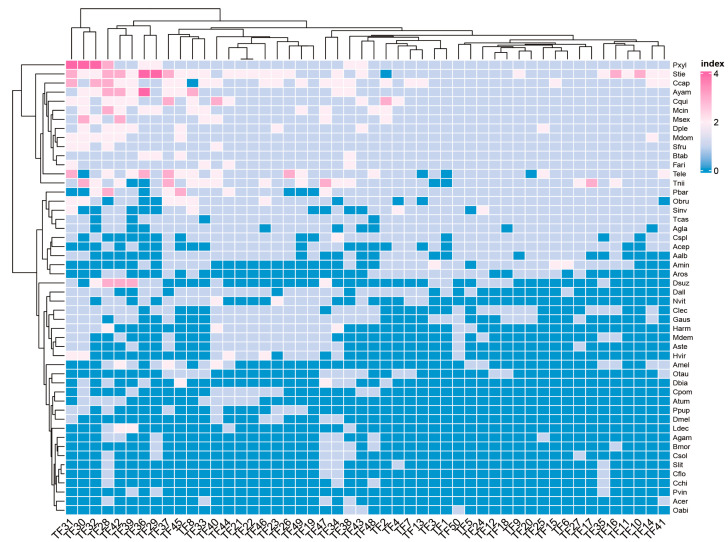
Characteristic heat map of NMF, the vertical axis is the name of insects, and the horizontal axis is the eigenvalues of different features. See Appendix A for the names of species assigned to each label.

**Figure 4 genes-14-00421-f004:**
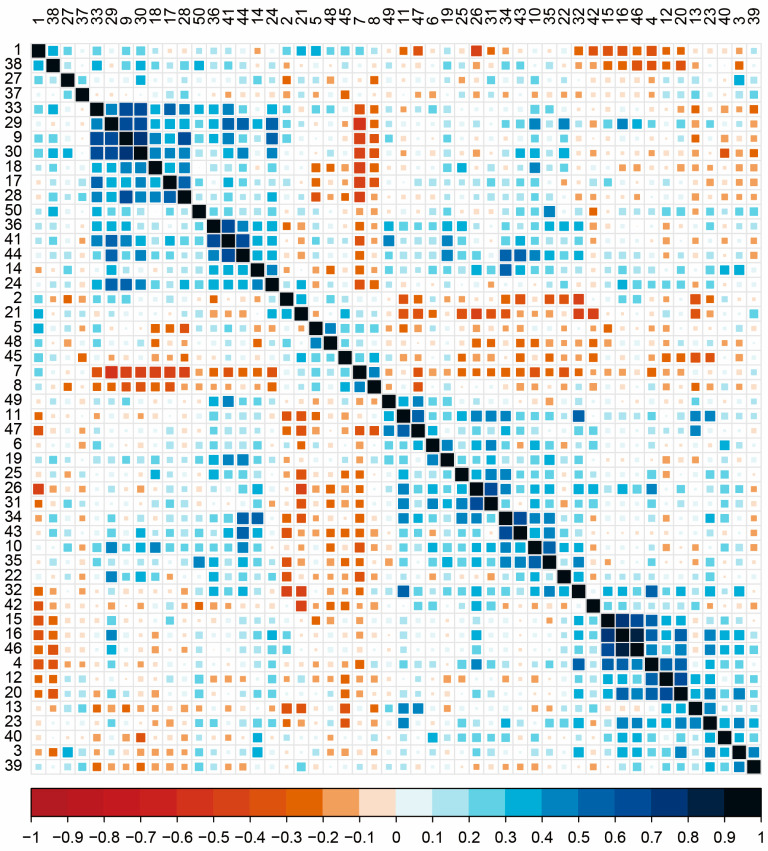
Heat map of the correlation between the statistical quantities of different gene families.

**Figure 5 genes-14-00421-f005:**
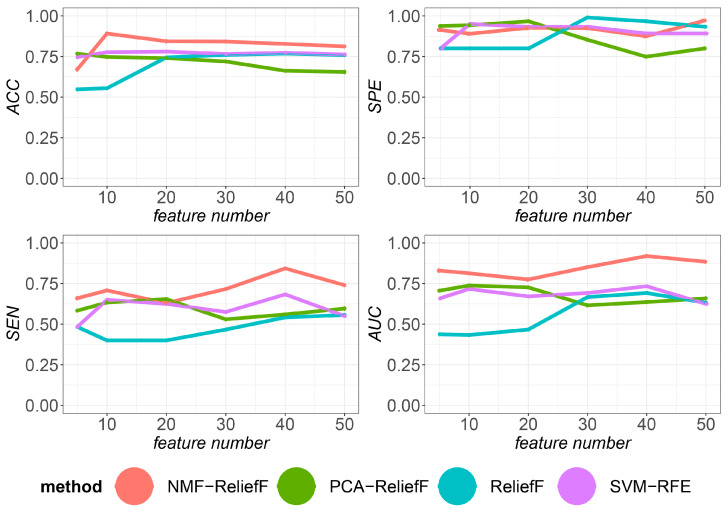
Evaluation index values for different algorithms at feature counts of 5, 10, 20, 30, 40, and 50, respectively. The four algorithms differed in accuracy, sensitivity, specificity, and *AUC* on the insect gene family data set.

**Table 1 genes-14-00421-t001:** Statistics of the microarray data sets.

Data Sets	Instance	Gene Number	Class	Disease
Gordon [48]	181	12,533	2	Lung Cancer
Tian [49]	173	12,625	2	Myeloma
Singh [50]	102	12,600	2	Prostate Cancer
West [51]	49	7129	2	Breast Cancer

**Table 2 genes-14-00421-t002:** Comparison of our method, ReliefF, SVM-RFE and PCA-ReliefF on high-dimensional microarray data sets; the best result is in bold face.

		Lung				Prostate				Myeloma				Breast			
Methods	Classifiers	ACC	SEN	SPE	AUC	ACC	SEN	SPE	AUC	ACC	SEN	SPE	AUC	ACC	SEN	SPN	AUC
ReliefF	k-NN	0.857	**0.990**	0.714	**0.866**	0.804	0.472	0.896	0.950	0.803	0.788	0.849	0.667	0.914	0.833	0.996	0.833
	RF	0.757	0.658	0.859	0.794	0.798	0.320	0.928	0.952	0.843	0.876	0.836	0.739	0.767	0.553	0.910	0.460
	SVM	0.847	0.864	0.858	0.880	0.809	0.129	**0.985**	**0.988**	0.883	0.888	0.904	0.801	0.904	**0.900**	**0.967**	**0.867**
SVM-RFE	k-NN	0.757	0.871	0.643	0.829	0.758	0.871	0.643	0.830	0.764	0.720	0.809	0.082	0.852	0.678	0.997	0.668
	RF	0.815	0.810	0.810	0.847	0.816	0.810	0.810	0.847	0.775	0.810	0.751	0.602	0.791	0.667	0.883	0.589
	SVM	0.847	0.860	0.883	0.849	0.848	**0.860**	0.883	0.850	0.892	0.880	0.906	0.799	0.910	0.867	0.950	0.833
PCA-ReliefF	k-NN	0.573	0.740	0.370	0.645	0.774	0.475	0.847	0.918	0.774	0.475	0.847	0.918	0.652	0.300	0.800	0.220
	RF	0.531	0.540	0.566	0.639	0.769	0.239	0.898	0.927	0.769	0.239	0.898	0.927	0.848	0.920	0.880	0.800
	SVM	0.546	0.560	0.550	0.602	0.878	0.581	0.956	0.980	0.878	0.581	0.956	0.980	0.850	0.960	0.927	0.807
NMF-ReliefF	k-NN	0.751	0.943	0.914	0.709	0.919	0.943	0.986	0.998	0.921	0.948	0.966	0.845	0.848	0.948	0.833	0.700
	RF	0.593	0.567	0.657	0.619	0.873	0.673	0.933	0.980	0.940	0.946	0.940	0.891	0.881	0.800	0.883	0.750
	SVM	**0.855**	0.846	**0.902**	0.843	**0.942**	0.833	0.978	0.985	**0.941**	**0.983**	**0.915**	**0.898**	**0.914**	0.800	0.933	0.800
Baseline	k-NN	0.545	0.783	0.425	0.550	0.734	0.325	0.836	0.883	0.682	0.688	0.739	0.509	0.557	0.400	0.653	0.317
	RF	0.545	0.575	0.508	0.600	0.774	0.233	0.931	0.956	0.717	0.783	0.713	0.553	0.467	0.417	0.503	0.200
	SVM	0.575	0.558	0.475	0.583	0.687	0.252	0.803	0.854	0.872	0.912	0.859	0.788	0.710	0.750	0.667	0.717

**Table 3 genes-14-00421-t003:** Comparison of our method, ReliefF, SVM-RFE, and PCA-ReliefF on matrix of gene family data set; the best result is in bold face.

Methods	Classifers	ACC	SEN	SPE	AUC	Time
ReliefF	k-NN	0.766	0.758	0.833	0.700	1.063
	RF	0.783	0.675	0.916	0.750	1.756
	SVM	0.786	0.541	0.966	0.691	1.074
SVM-RFE	k-NN	0.770	0.708	0.866	0.725	3.396
	RF	0.730	0.675	0.825	0.708	4.057
	SVM	0.786	0.683	0.891	0.733	3.492
PCA-ReliefF	k-NN	0.669	0.573	0.749	0.653	0.060
	RF	0.609	0.526	0.609	0.609	0.644
	SVM	0.667	0.560	0.744	0.636	0.066
NMF-ReliefF	k-NN	0.745	0.443	0.370	0.788	1.310
	RF	0.723	0.696	0.765	0.800	2.038
	SVM	**0.843**	**0.843**	**0.974**	**0.919**	**1.324**
Baseline	k-NN	0.643	0.750	0.725	0.629	0.069
	RF	0.663	0.600	0.650	0.587	9.657
	SVM	0.683	0.566	0.783	0.629	0.097

## Data Availability

Insect genome sequences and annotation files are available at The National Center for Biotechnology Information (https://www.ncbi.nlm.nih.gov/genome, accessed on 1 February 2022), InsectBase (http://v2.insect-genome.com/Genome, accessed on 15 March 2022), VectorBase (https://vectorbase.org/vectorbase/app, accessed on 22 March 2022), Fireflybase (http://www.fireflybase.org/jbrowse, accessed on 23 March 2022), Ensembl Genomes (https://metazoa.ensembl.org/index.html, accessed on 24 March 2022), and GigaDB (http://gigadb.org/dataset/100001, accessed on 24 March 2022). The microarray data are available at GEO (https://www.ncbi.nlm.nih.gov/geo, accessed on 1 July 2022).

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
