# Peer review of "An Efficient Feature Selection Algorithm for Gene Families Using NMF and ReliefF"

_genes, 2023, doi:10.3390/genes14020421_

Round 1
Reviewer 1 Report
1. The gap, i.e., the limitation of the current tool (TreeFam needs to be elaborated in the Introduction). The motivation of the study also needs to be clarified in the abstract. What characteristics of a gene family that have not been statistically described in the abstract?
2. Line 76-78 is not clear
3. What are the criteria for selecting the four cancer microarray datasets in evaluating the performance of the proposed technique?
4. What is the justification for choosing the 139 genome sequences in the study followed by 50 insects? What are the criteria for selecting 50 insects for further analysis? The information needs to be added in the Materials and Methods section. The overall process needs to be prepared in a Figure for easier understanding.
5. Grammatical error: the algos, i.e., NMF-ReliefF, needs to be written similarly throughout the manuscript.
Reviewer 2 Report
In this paper, the authors present a feature selection approach for extracting species traits at the gene family level. The authors develop a new Classification algorithm by combining ReliefF algorithm and non-negative matrix decomposition (NMF). Their approach improves over the inefficiencies of existing methods. Experimental evaluation confirms the robustness and accuracy of their approach.
The reviewer believes that English presentation can be greatly improved. The authors should critically review the entire paper to improve the English presentation. Following is a sample of sentences that should be rewritten to improve presentation, readability, and thus impact of the paper.
First, the title, “An efficient feature selection algorithm for gene families using
NMF and ReliefF” reads better than the current title.
Page 1.
1. However, the characteristics of gene families have not been well statistically and summarized.
2. In evolution, gene families have expanded and contracted, are caused by a variety of factors, including natural selection, genetic drift, and gene duplication.
Page 2.
1. Its goal is to provide a curated database that provides an accurate evolutionary history of all animal gene families and trustworthy ortholog and paralog assignments.
2. classifying using statistics' high-dimensional matrix remains tough.
Page 3
1. Since there are several alternative splicing in genomic mRNA sequences, and some genes are produced differently by distinct exon splicing techniques, alternative splicing is removed from the genomic mRNA sequence.
2. This script will collect the number of each TreeFam gene family for 119 each species, producing a matrix data set. Figure 1 illustrates the workflow for the pro- 120 posed predicting methods. (Note: Infact entire first para of Section 2. Materials and Methods can be improved)
3. The proposed framework was mainly performed on the MATLAB R2018A platform.
Page 4.
1. The reviewer sees no value of first para of the Section 2.3. feature selection, expect may be the last sentence that should be rewritten to improve it.
Page 11.
1. Due to the difficulty in determining the ideal number of selected genes, the classification performance of multiple approaches employing varying numbers of selected genes is tested here.
